# Characterization of a *Bacillus velezensis* with Antibacterial Activity and Its Inhibitory Effect on Gray Mold Germ

**Lei Li [1], Rongjie Wang [2], Xingxing Liang [3], Yunpeng Gai [3] , Chen Jiao [4] and Meiqin Wang [1],***

1   College of Plant Protection, Shanxi Agricultural University, Jinzhong 030801, China; qstlilei@stu.scau.edu.cn
2   College of Plant Protection, China Agricultural University, Beijing 100193, China; sy20223193277@cau.edu.cn
3   School of Grassland Science, Beijing Forestry University, Beijing 100083, China; liangxingx@bjfu.edu.cn (X.L.);
    gaiyunpeng@bjfu.edu.cn (Y.G.)
4   Key Laboratory of Molecular Biology of Crop Pathogens and Insects, Institute of Biotechnology,
    Zhejiang University, Hangzhou 310058, China; biochenjiao@zju.edu.cn
*   Correspondence: mqwang@sxau.edu.cn

**Abstract:** The present study provides a comprehensive overview of the *Bacillus velezensis* strain Htq6, and its potential applications in plant disease control. Htq6 is an endophytic bacterium derived from walnut, which was found to possess a strong inhibitory effect on a wide range of plant pathogenic microorganisms and was identified as a good plant disease control agent. The entire genome of the *Bacillus velezensis* Htq6 was sequenced, and a comparative genomic analysis was conducted with various *Bacillus* species in order to better understand the mechanism of the strain's biological control. At the same time, a new classification result was presented. Additionally, transcriptome analysis was performed to explore the response mechanism of tomato gray mold fungus after treatment with the fermentation liquid of *Bacillus velezensis* Htq6. The study analyzed the distribution of various secondary metabolite gene clusters in the *Bacillus* model strains and employed RNA-Seq technology to obtain transcriptome expression profiles. Furthermore, the cell wall, cell membrane, and antioxidant-related genes of *Botrytis cinerea* were analyzed, providing insight into the antibacterial mechanism of biocontrol bacteria and the stress response mechanism of *Botrytis cinerea*. The results of the research are promising, and could potentially lead to the development of an effective biocontrol agent for the prevention and control of various plant diseases.

**Keywords:** *Bacillus velezensis*; genome sequencing; comparative genome; gray mold; transcriptome

## 1. Introduction

Biocontrol agents are living organisms (e.g., bacteria, fungi, viruses, and insects) that are employed to regulate the growth of pests and pathogenic microorganisms [1,2]. These agents are used to create biopesticides, which alter the environment of the target pests, resulting in the reduction of plant diseases and insect pests. The biopesticides generated by these agents are often less hazardous than chemical pesticides, making them beneficial for the environment and human health [3]. Additionally, their application may reduce or even eliminate the use of chemical pesticides, thus decreasing the cost of agricultural production [4]. Moreover, biocontrol agents are more specific and selective, thus effectively reducing the damage caused by pests and pathogens. The application of these agents (e.g., Topshield, Trichodex, ContansWG, SoilGard, and Endothin parasiticac) has been proven to be a safe, efficient, and economical approach for managing plant diseases and insect pests [5]. Although the use of agents for pest and pathogen control is a promising approach, it is still in the early stages of development and certain obstacles need to be overcome [6]. One issue is that the agents are typically tailored to a particular species, which limits the range of organisms that can be targeted. Another challenge is the cost associated with utilizing these agents [7]. Thirdly, their application may require more time and labor than that of chemical pesticides, thus increasing the cost of labor. Finally, they may not be

as effective as chemical pesticides in certain cases [8]. Despite these challenges, biocontrol agents are a safe, effective, and economical method for managing plant diseases and insect pests. Therefore, further research is needed to tackle the issues of cost, effectiveness, and species selectivity. To ensure the successful application of these agents, the development of new methods for producing biopesticides, and the development of new biocontrol agents should be considered. Additionally, the government should provide more financial and technical support for the research and development of biocontrol agents [9]. With the advancement of science and technology, biocontrol agents will become more widely used, and they will play an essential role in the sustainable development of agriculture and the protection of human health.

The discovery and study of *Bacillus* is of immense importance for future control of plant diseases and human illnesses. Its potential uses in food, medicine, agriculture, and environmental protection are being constantly explored and expanded, thus making it a promising method to replace the use of chemical pesticides [10,11]. Moreover, understanding the mechanisms of action of *Bacillus*, its effectiveness in controlling various plant diseases, and its potential side effects on the environment is essential in order to fully exploit its potential. In addition, exploring the application of other biocontrol agents such as fungi, viruses, and bacteria in agricultural production, finding a suitable combination of biocontrol agents for different diseases and environments, and realizing the sustainable development of agriculture are also important objectives [12]. Moreover, biocontrol agents can also be used in other fields such as food processing, environmental protection, and medicine. The application of biocontrol agents in medicine can not only prevent and control human diseases, but also reduce the use of chemical drugs, thus providing long-term benefits to people's health. Furthermore, the application of biocontrol agents in various fields can promote the development of green technology and contribute to the sustainable development of society [13–15].

The production of antibiotic molecules by *Bacillus subtilis*, *Bacillus amyloliquefaciens*, *Bacillus velezensis*, and *Bacillus siamensis* species is an important contribution to their ecological niche [16,17]. These molecules are derived from two major pathways: the ribosomal pathway, which produces bacteriocins, and the non-ribosomal pathway, which produces lipopeptides, polyketones, and other compounds [18]. Bacillomycin was first isolated from *Bacillus* in 1948 and then different types of molecules, such as Bacillomycin, surfactin, Iturin, and fengycin, have since been identified and their structures have been determined [19,20]. This wide breadth of structures provides these molecules with a wide range of antibacterial activities, allowing them to effectively address diverse environmental niches. Moreover, the diversity of structures and compositions of these active substances leads to their important functions in various environmental niches. Therefore, the production of antibiotic molecules by these four species of *Bacillus* provides an important contribution to their ecological niche.

*Bacillus* strains are known to exhibit unique metabolic profiles and different antimicrobial abilities, making it important to develop accurate predictive strategies [21–23]. In order to gain a better understanding of the metabolic and resistance abilities of *Bacillus*, genomic and biosynthetic gene cluster studies of *Bacillus* have been conducted. The comprehensive analysis of the entire genome of different *Bacillus* species and the synthesis gene clusters of some important antibacterial substances has helped to identify the specific genes involved in the production of antibacterial substances, which may be used as biocontrol agents [24–26]. Furthermore, the comparison of different biosynthetic gene clusters has provided insights into how these strains can evolve different antibacterial profiles and resistance abilities [27]. Ultimately, this research has advanced our understanding of the metabolic and resistance capabilities of *Bacillus* and has implications for the development of new biocontrol agents and the study of resistance mechanisms.

*Botrytis cinerea* is a plant pathogen that has caused considerable devastating diseases and economic losses to horticultural crops. More than two hundred species have been listed as its hosts, including vegetables, fruit trees, and horticultural ornamentals [28].

Gray mold of tomato is one of the main fungal diseases that harm tomato. The annual yield loss caused by gray mold is more than 20%, and the economic losses run into tens of billions of dollars. In China, tomato is planted on a large scale and has concentrated planting areas, which makes the occurrence of disease an important problem faced by facility vegetable planting [29]. The use of chemical means to control the ecological balance of the environment and human health had a greater potential harm, so the biological control strategy is widely promoted [30]. *Bacillus velezensis* Htq6, isolated from walnut fruit, can be successfully colonized in tomato [31]. Furthermore, a wettable powder has been preliminarily developed by screening various additives, which has produced good control effects on tomato gray mold in the field, although its exact mechanism of action has yet to be determined.

Here, we present the entire genome sequence of *Bacillus velezensis* Htq6, a novel strain that was isolated from walnut fruit. We conducted whole-genome sequencing to determine the genome size, GC ratio, position and function of coding genes, as well as secondary metabolites that could be synthesized. Additionally, we compared the phylogenetic topology and biosynthetic gene clusters (BGCs) with other *Bacillus* spp. to identify any special antibacterial substances that have potential as biocontrol agents. To further explore the regulatory mechanism of Htq6 and the response mechanism of *Botrytis cinerea*, we also carried out transcriptome sequencing on *Botrytis cinerea* treated with Htq6 fermentation broth, and analyzed the pathways that were significantly affected.

## 2. Materials and Methods

### 2.1. Bacterial Strains, Growth Conditions and DNA Extraction

The *Bacillus velezensis* Htq6 strain was isolated from walnut fruit from Shanxi Province, China. A single colony was then inoculated into 100 mL of liquid LB medium (consisting of 1 g of tryptone, 1 g of peptone, and 0.5 g of yeast extract) at 180 rpm and 27 °C for 2 days [31,32]. Subsequently, a part of the bacterial solution was preserved in a $-80$ °C fridge in glycerol/water mixtures (30 vol % glycerol), while another part was used for genomic DNA extraction, following the steps of the extraction kit (provided by Sangon Biotech, Shanghai, China) carefully. Finally, the quality and concentration of the genomic DNA were tested using agarose gel electrophoresis and a NanoDrop. Prior to the experiment, the medium and solution had been sterilized at 121 °C for 20 min.

### 2.2. Genome Sequencing and Phylogenetic Analysis

The genome was sequenced using the Illumina HiSeq 2000 at Beijing Anoroad. After quality filtering, high-quality paired-end reads were obtained and then assembled using SOAPdenovo v.2.04 to generate a draft genome [33]. Subsequently, the genome was submitted to the NCBI Prokaryotic Genomes Automatic Annotation Pipeline for annotation.

Recently, it was suggested that *Bacillus amyloliquefaciens*, *Bacillus siamensis*, and *Bacillus velezensis* were closely related and formed an "operational group *Bacillus amyloliquefaciens*". In order to investigate this hypothesis, a total of 333 genome sequences of prokaryotes were retrieved from the GenBank database. Specifically, 84 strains of *Bacillus amyloliquefaciens*, 7 strains of *Bacillus siamensis*, 242 strains of *Bacillus velezensis*, and 1 strain of *Bacillus subtilis*, with *Bacillus siamensis* XY18, *Bacillus amyloliquefaciens* DSM7, *Bacillus velezensis* FZB42, and *Bacillus subtilis* 168 as the type strain, respectively. Subsequently, gyrA, gyrB, and rpoB gene sequences were downloaded from the NCBI nucleotide resource database for phylogenetic analysis. MEGA6 was used to construct a phylogenetic tree using the maximum likelihood method with *Bacillus subtilis* 168 set as the outgroup. Additionally, the average nucleotide identity (ANI) value of the genomes and protein sequence similarity were calculated based on the MUMmer alignment method [34,35].

### 2.3. Comparative Analysis of Gene Clusters for Antimicrobial Compounds

The BGCs for secondary metabolites were predicted by utilizing the bacterial mode of antiSMASH (v5.0.0) [36], a program designed for the identification and analysis of

secondary metabolite gene clusters. Subsequently, MUMmer was employed to align all secondary metabolite gene clusters, and any clusters that showed a sequence similarity of 70% or higher and a sequence coverage of 5% or greater were then classified into one category. Finally, the gene clusters were graphically represented using genoPlotR, a program that provides visualizations of gene clusters and operons.

### 2.4. RNA Extraction and Transcriptome Analysis

The *Bacillus velezensis* Htq6 strain was inoculated into 100 mL of liquid LB medium at 180 rpm and 27 °C for 2 days, before being filtered through a 0.22 µm bacterial filter to remove any bacterial cells, in order to obtain a sterile fermentation broth. Subsequently, the preserved *Botrytis cinerea* spores were inoculated onto PDA medium plates, and the concentration of spores was adjusted to $1 \times 10^4$/mL. A 1 mL volume of the *Botrytis cinerea* spore suspension was then inoculated into 100 mL of an optimized medium, which was then shaken in a culture for 3 to 5 days at 180 rpm and 24 °C. For the treatment group, *Bacillus velezensis* Htq6 fermentation broth was added to attain a final concentration of 10%, while the control group had no substance added. After oscillating the culture for 12 h, the total RNA was extracted using the Axygen RNA purification kit (Capital Scientific, Austin, TX, USA) according to the manufacturer's instructions. RNA integrity and concentration were then assessed and evaluated using the RNA Nano 6000 Assay Kit of the Bioanalyzer 2100 system (Agilent Technologies, Santa Clara, CA, USA).

RNA-Seq was performed using three biological replicates of each sample. The libraries were constructed using the Ultra RNA Library Prep kit (NEB) and were sequenced using an Illumina HiSeq 2000 sequencer platform (Illumina Inc., San Diego, CA, USA) to generate 150 bp paired-end reads. Subsequently, Trimmomatic V0.36 was employed to process the raw RNA-Seq readings in order to eliminate junctions and low-quality bases [37]. In addition, HISAT2 version 2.1.0 was utilized to map the cleaned reads to the *Botrytis cinerea* [38]. Afterward, HTSeq was applied to count the number of reads that were mapped to each gene [39], and the RDESeq package was utilized to perform differential expression analysis with the preset parameters of FDR $\leq$ 0.05 and Log2FC $\geq$ 2 [40]. To further annotate the DEGs, Blast2GO software was employed for gene ontology (GO) annotation and functional term mapping [41]. Moreover, the DEGs were classified into GO categories using Web Gene Ontology Annotation Plot 2.0 (WEGO) [42]. Lastly, KEGG enrichment analysis was carried out after obtaining KEGG Ortholog annotation results using KAAS (https://www.genome.jp/tools/kaas/ accessed on 28 March 2023).

### 2.5. RT-qPCR Analysis

To further verify the RNA-Seq transcriptome data, the Applied Biosystems QuantStudioTM 6 Flex qPCR platform was applied to perform quantitative reverse transcription PCR (qRT-PCR). The RNA extraction methods employed were the same as previously described protocols. A comprehensive list of all the primers employed in the RT-qPCR analysis can be found in Supplementary Table S1.

## 3. Results

### 3.1. Species Assignment of the Strain Htq6

The 16S rRNA and gyrA and gyrB gene sequences of the Htq6 strain were obtained from its chromosomal DNA by PCR amplification employing universal oligonucleotide primers followed by sequenced as described by Lee et al. [43]. The sequencing results were subsequently analyzed by BLAST against the GenBank database which indicated that the Htq6 strain had a high level of similarity (99% or 95%) to *Bacillus velezensis* and *Bacillus amyloliquefaciens*. To further assess this phylogenetic relationship, strains with gyrA and gyrB gene sequence similarities of more than 95% were used. A phylogenetic tree was then constructed based on the concatenated gyrA and gyrB sequences of related members of the *Bacillus* genus, which revealed that strain Htq6 should be assigned to *Bacillus velezensis* rather than *Bacillus amyloliquefaciens*.

### 3.2. Features of Bacillus velezensis Htq6 Genome

The complete genome of the strain Htq6 was assembled, consisting of 3,888,123 base pairs with a GC content of 46.54% (Figure 1). Gene prediction revealed that there were 3826 protein-coding genes (Table S3), which were classified into 22 functional categories of Clusters of Orthologous Groups (COGs). The largest category was general function prediction only (671 genes), followed by genes related to amino acid transport and metabolism (528 genes) (Figure 2). Additionally, 146 genes related to carbohydrate active enzymes (CAZymes) were identified, including 46 glycoside hydrolases (GHs), 38 glycosyltransferases (GTs), 3 polysaccharide lyases (PLs), 32 carbohydrate esterases (CEs), 7 auxiliary activities (AAs), and 20 carbohydrate-binding related enzymes (CBMs). These CAZymes are very important proteins in plant pathogenic fungi and bacteria and play a vital role in their growth and development. As a biocontrol strain, Htq6 can not only produce antibiotics, antagonistic proteins, or peptides, but also secrete enzymes which act on the corresponding substrates on the fungal cell wall, thus limiting the expansion and reproduction of pathogens by dissolving the bacteria. Moreover, many carbohydrate enzyme families can synthesize chitinase, glucanase, cellulase, xylanase, pectinase, protease, etc. Not only does this promote the growth and reproduction control of bacteria that are antagonistic to pathogens, but it also plays an important role in their competition in terms of nutrients and ecological sites.

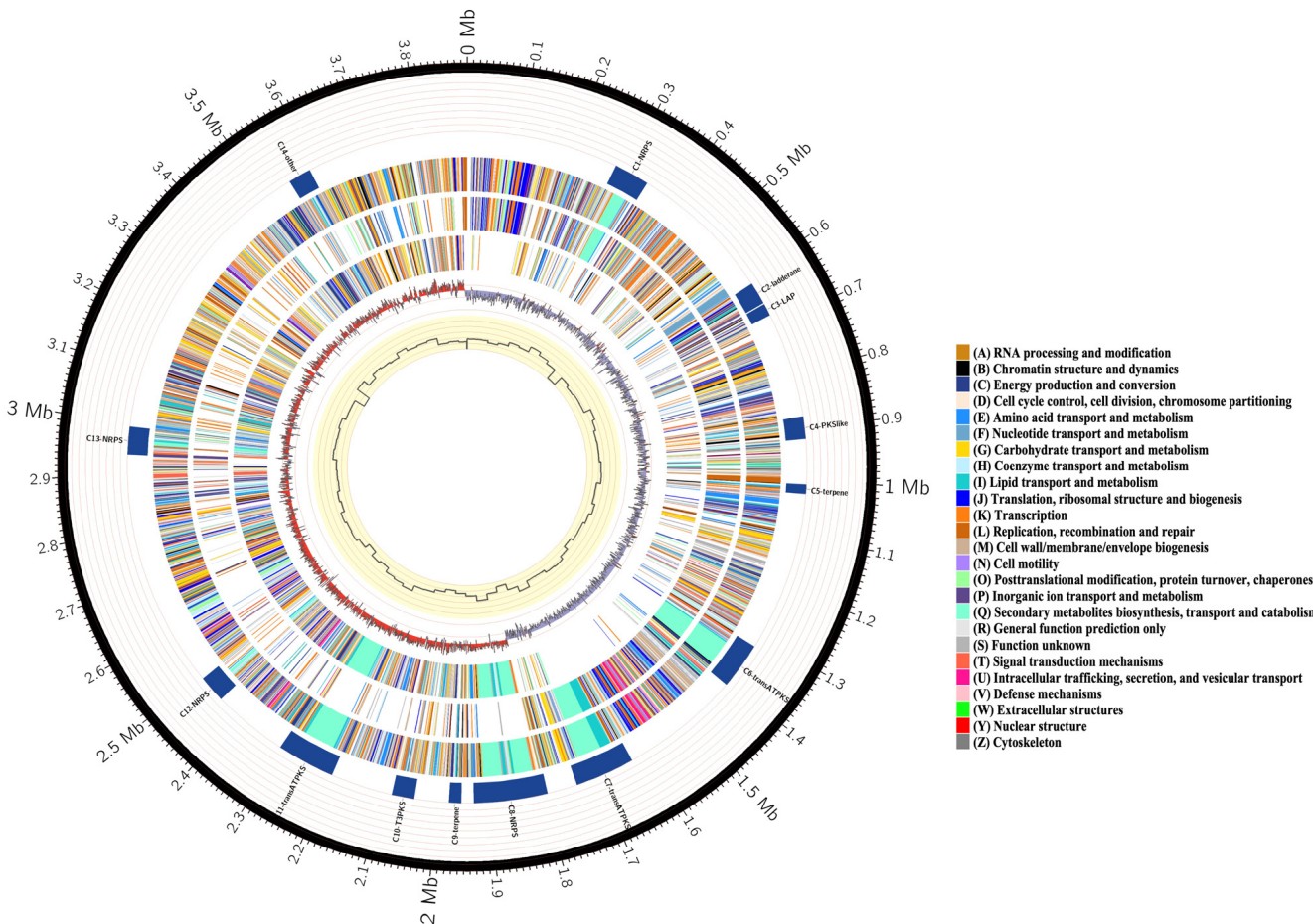

**Figure 1.** Genome map of *Bacillus velezensis* strain Htq6. The seven circles (outer to inner) represent strand genomic sequence, locations of predictive clusters, strand CDSs, sense strand CDSs, antisense strand CDSs, GC skew, and GC content.

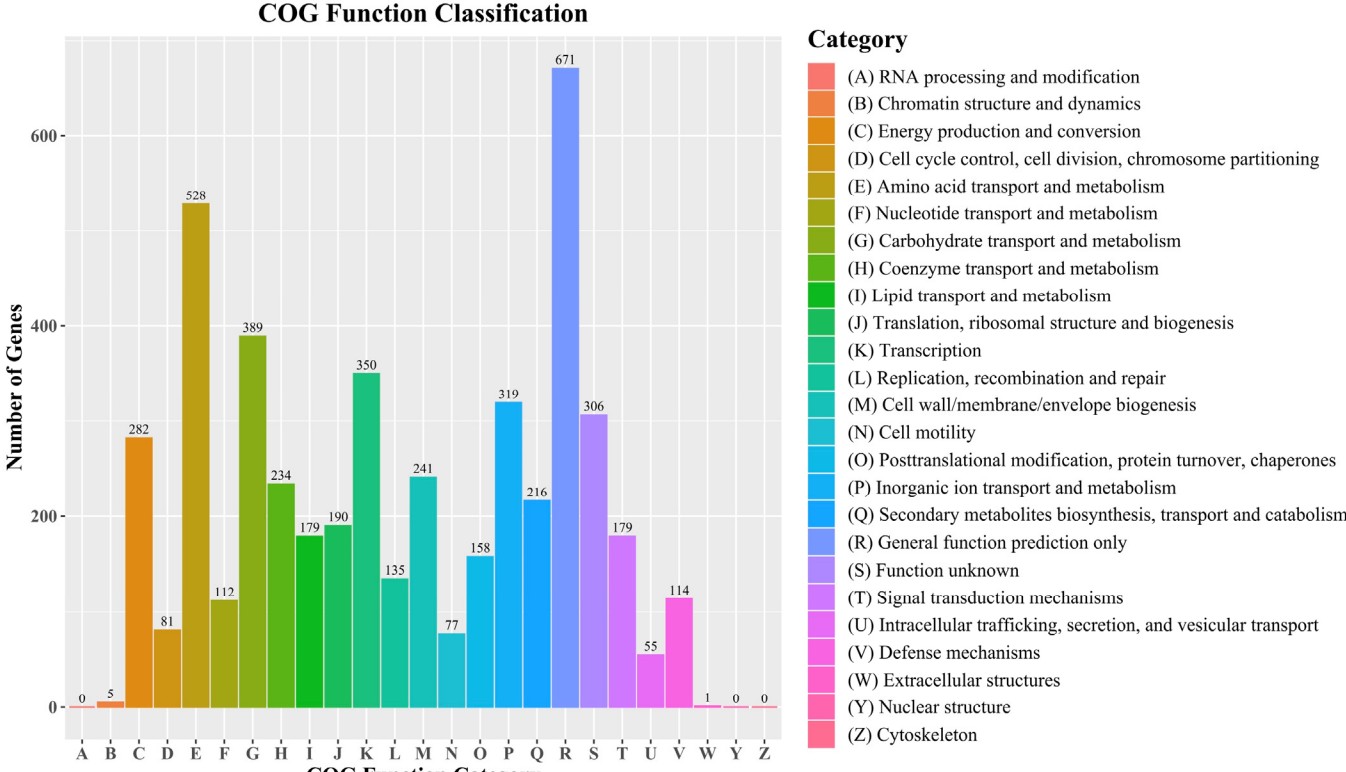

**Figure 2.** Gene distribution based on Clusters of Orthologous Groups of *Bacillus velezensis* Htq6. The horizontal coordinates are the function classes of COG, and the vertical coordinates are the numbers of unigenes in each class. The notation on the right is the full name of the functions on the *x*-axis.

AntiSMASH was used to analyze the genome sequence of strain Htq6 and predicted a total of 14 biosynthetic gene clusters with high potential for producing secondary metabolites (Figure 3). Notably, several of the predicted gene clusters showed strong homology with the reference clusters, including plantazolicin, macrolactin, bacillaene, engycin, difficidin, bacillibactin, and bacilysin. Moreover, the analysis revealed the potential of biocontrol bacteria to produce various secondary metabolites, which may play an important role in navigating the tight living space, preemptively colonizing host plants, and promoting synergistic symbiosis.

### 3.3. Phylogenetic Analysis of Bacillus amyloliquefaciens, Bacillus siamensis, and Bacillus velezensis

There are different views regarding whether *Bacillus velezensis* is a later heterotypic synonym of *Bacillus amyloliquefaciens*. In order to gain further insight into the degree of relatedness between them, we studied protein-coding genes with higher genetic diversity to classify and identify closely related taxonomic groups and ecotypes. The extremely conserved gyrA/gyrB gene encodes the DNA gyrase subunit A/B, while the rpoB gene encodes the DNA-dependent RNA polymerase β-subunit. Our phylogenetic tree constructed based on the similarity of conserved protein sequences revealed that with *Bacillus subtilis* 168 as the outgroup, three main monophyletic clades were corroborated by 100% bootstrap values (Figure 4). Clade I comprised 23 strains, with *Bacillus amyloliquefaciens* DSM7 ATCC233 as the type strain; Clade II was composed of 7 *Bacillus siamensis* strains based on the type strain XY18; Clade III contained Htq6, 243 *Bacillus velezensis* and 58 *Bacillus amyloliquefaciens* strains, with the type strain of *Bacillus amyloliquefaciens* FZB42 included in this Clade (Supplementary Table S2). Furthermore, phylogenetic analysis was conducted by calculating the average nucleotide identity (ANI) value of the genome and gyrA, gyrB, and rpoB gene sequences, and the results once again verified the accuracy of using the conserved genes

(gyrA/gyrB) for classification (Figures 5 and 6). The only strain showing discrepancy in the phylogenetic analysis based on the rpoB gene was *Bacillus amyloliquefaciens* DH8030. Therefore, based on the gyrA or gyrB gene, the three *Bacillus* species can be accurately distinguished.

### 3.4. The Species of Biosynthetic Gene Cluster Were Compared and Analyzed

Biocontrol *Bacillus* have an enormous capacity to synthesize secondary metabolites. Through a collinearity comparison analysis of 36 *Bacillus* strains, the secondary metabolite gene clusters were divided into 35 categories (Figure 7). Compared with *Bacillus amyloliquefaciens*, *Bacillus velezensis* had more types of secondary metabolite gene clusters. Among the 35 secondary metabolite gene clusters, group 1, group 2, group 4, group 5, group 6, and group 7 were gene clusters that were shared by both *Bacillus amyloliquefaciens* and *Bacillus velezensis*. *Bacillus amyloliquefaciens* DSM7 was used as the reference (Figure 8), with cluster 2 (lanthipeptide) being an antibacterial gene cluster that was shared by most of *Bacillus amyloliquefaciens*. Taking Htq6 as the reference strain (Figure 9), cluster 11 (transAT PKS like) was a specific gene cluster type of *Bacillus velezensis*, which is capable of producing difficidin, an antibacterial substance. Even though was only *Bacillus amyloliquefaciens* YP6 and *Bacillus amyloliquefaciens* MT45 possessed cluster 6 (transAT PKS), it was present in all *Bacillus velezensis* and was capable of producing the antibacterial substance macrolactin.

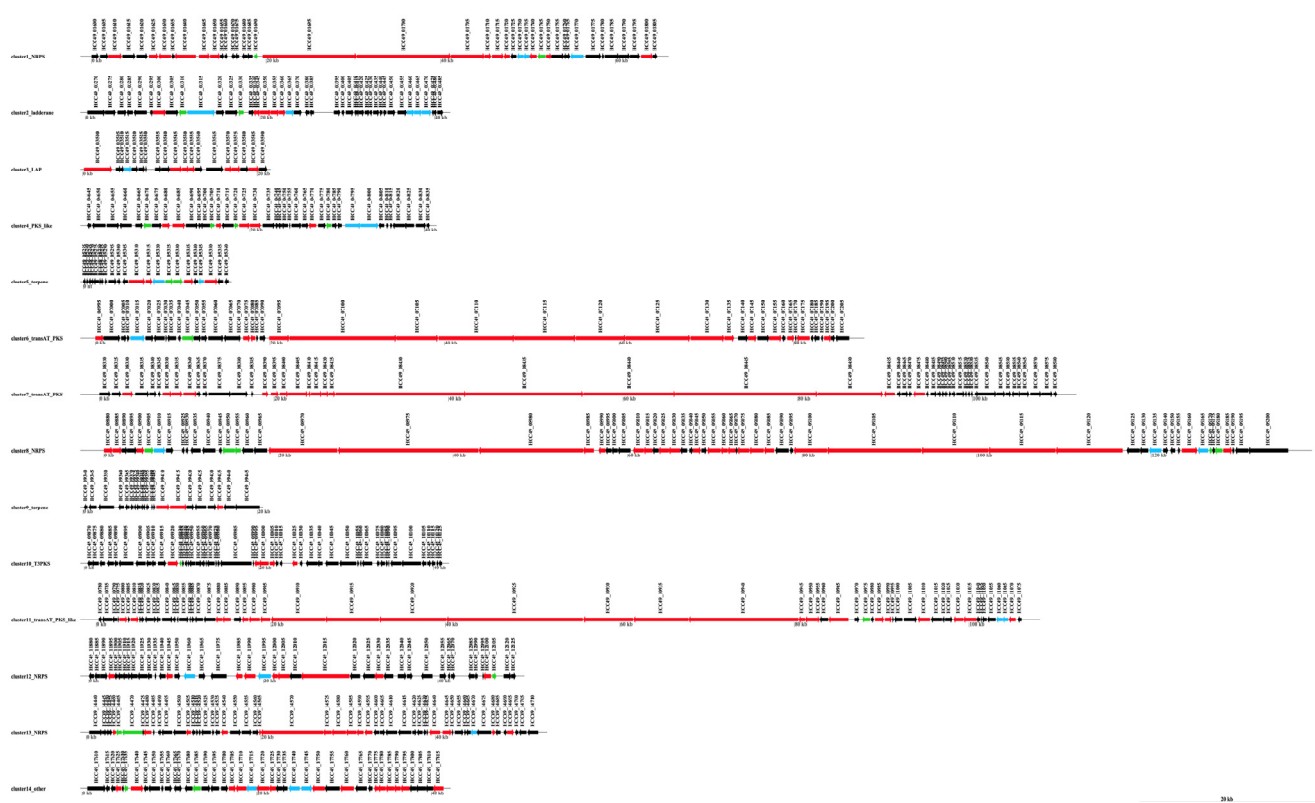

**Figure 3.** The secondary metabolite gene clusters of *Bacillus velezensis* Htq6.

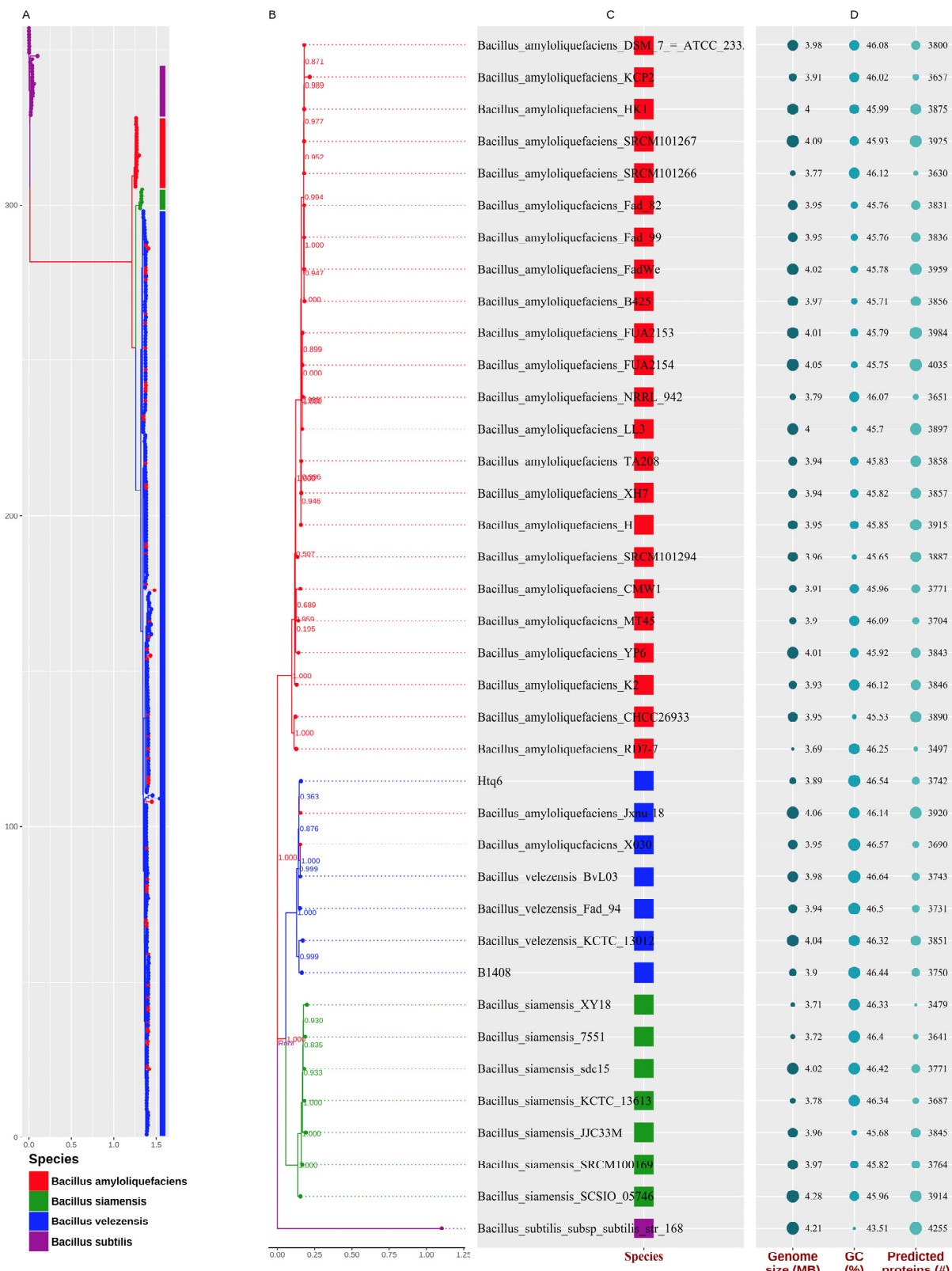

**Figure 4.** Phylogenetic analysis constructed using MAFFT and FastTree based on the genomic conserved protein sequences. (**A**) The phylogenetic analysis of conserved proteins in a total of 333 strains with *Bacillus subtilis* 168 as outgroup. (**B**) Taking 168 as an outgroup, the phylogenetic tree of conserved proteins of some strains including Htq6 was shown. (**C**) Strain name and classification (**D**) Three bubble plots illustrating genome size, GC (%), and number of predicted proteins.

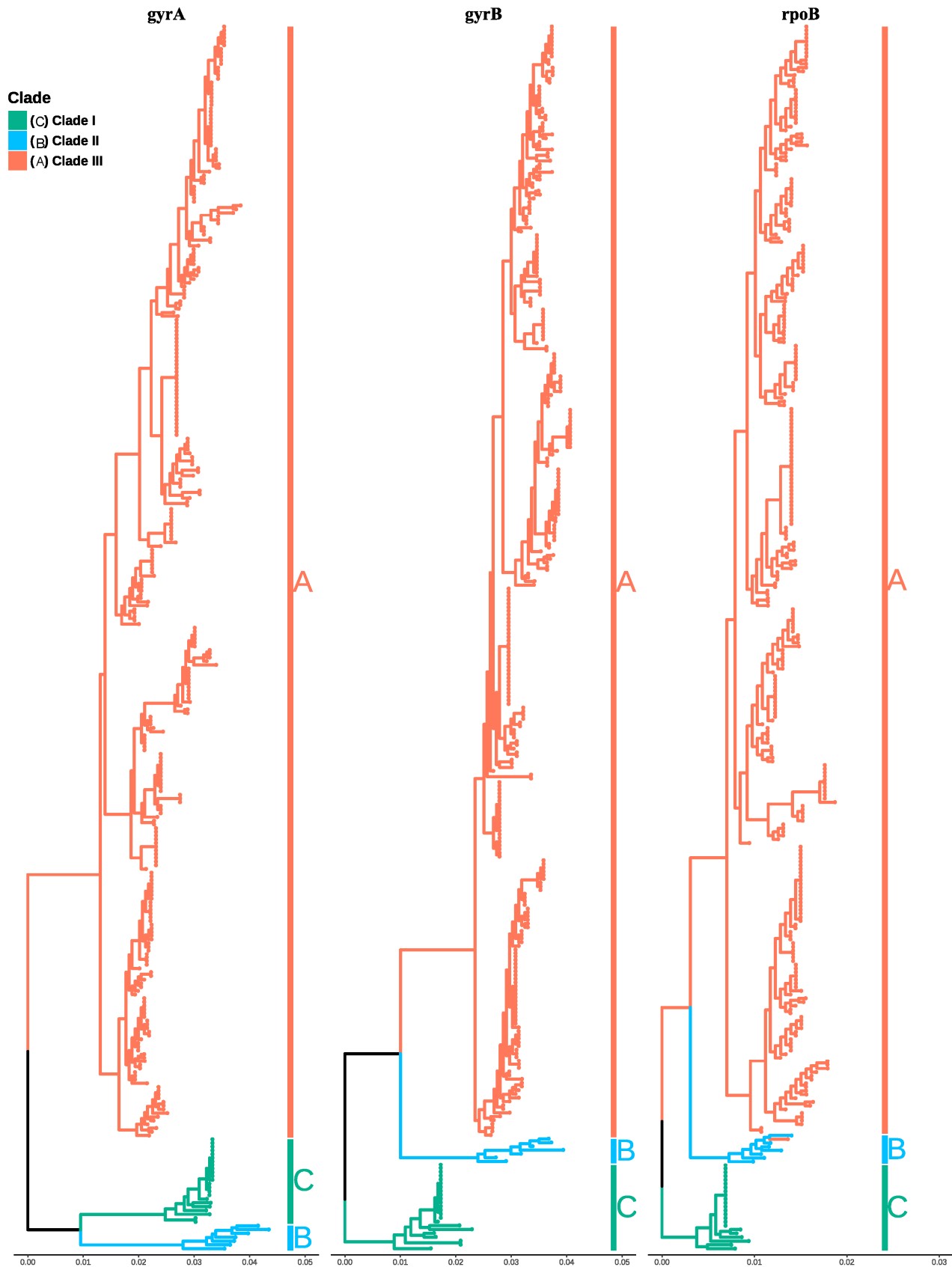

**Figure 5.** Phylogenetic analysis of *Bacillus amyloliquefaciens*, *Bacillus velezensis*, and *Bacillus siamensis* constructed using MEGA v6 based on gyrA, gyrB, and rpoB gene sequences.

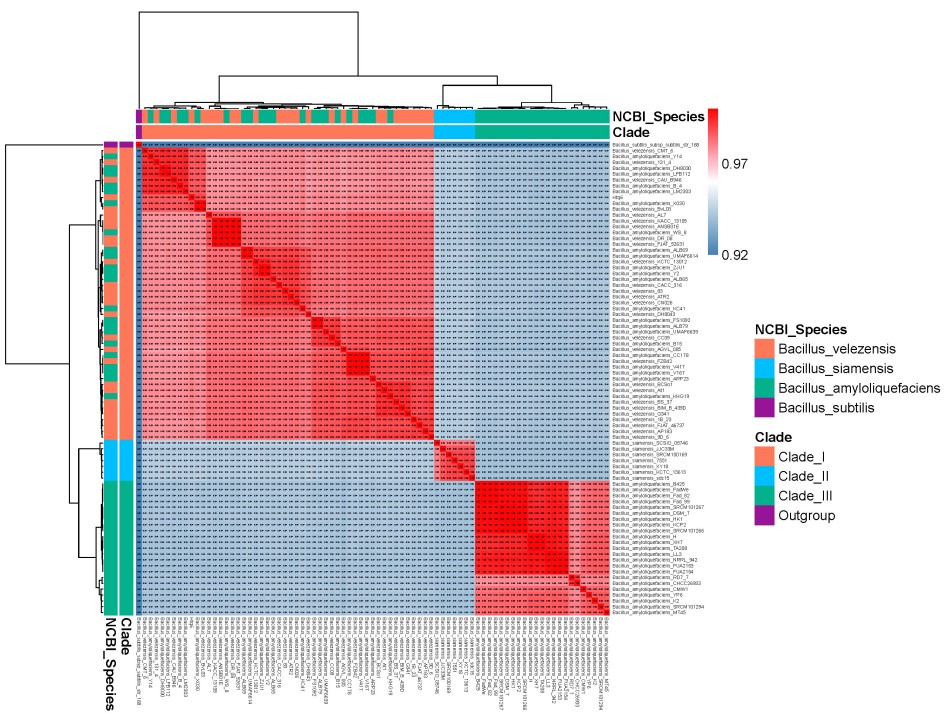

**Figure 6.** Average nucleotide identity (ANI) heat map analysis of different *Bacillus* species.

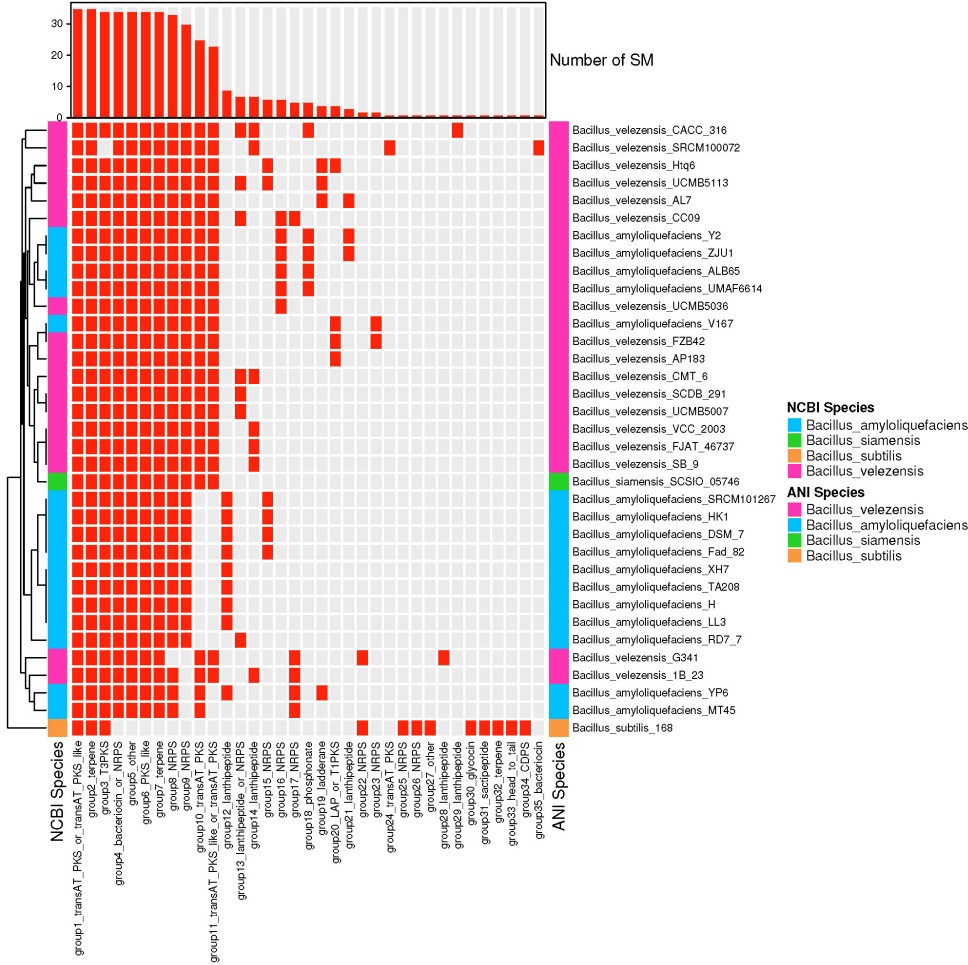

**Figure 7.** Distribution of different secondary metabolite gene clusters in *Bacillus* strains.

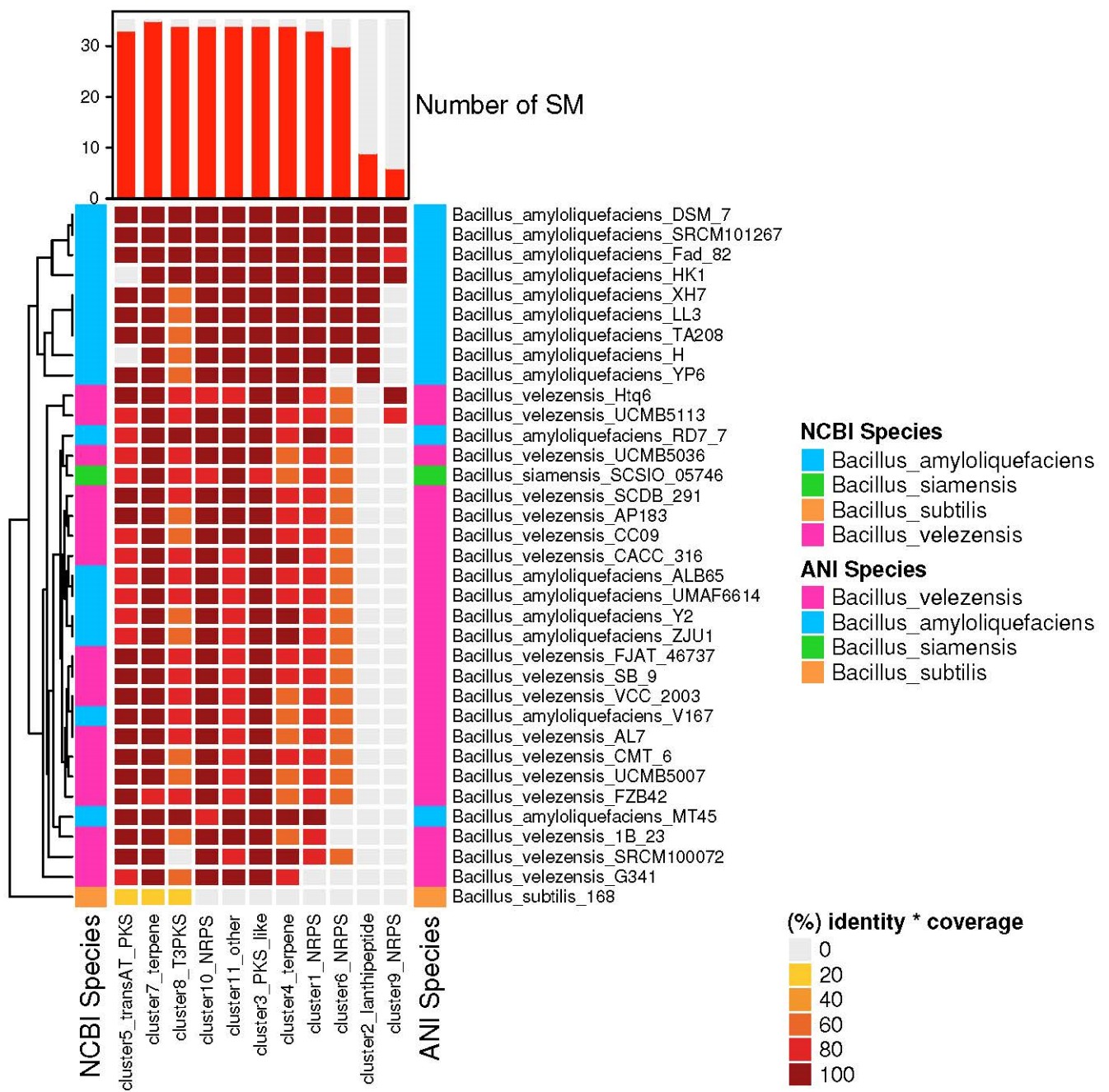

**Figure 8.** Comparison of secondary metabolite gene clusters with *Bacillus amyloliquefaciens* DSM7 as the reference strain.

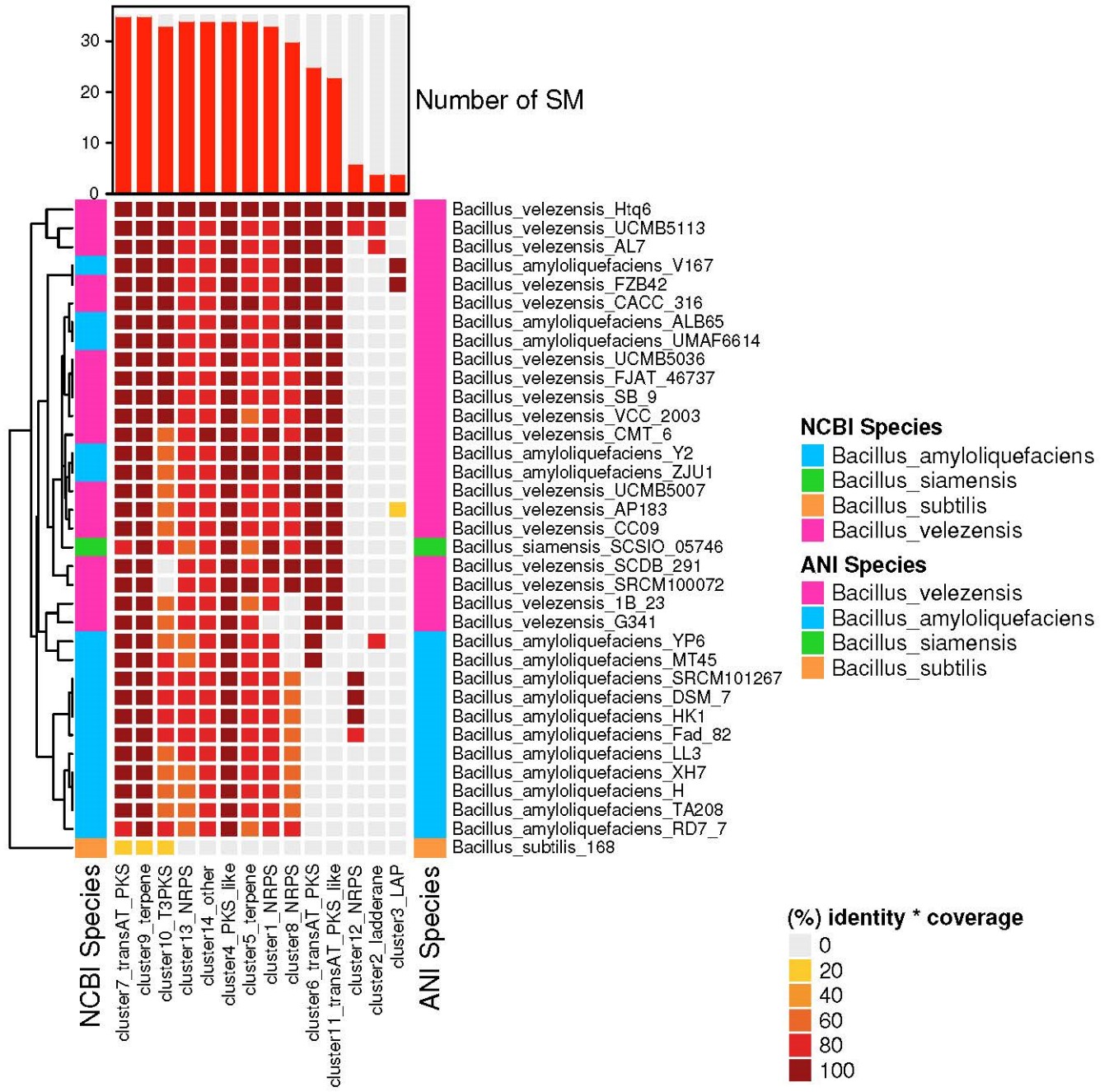

**Figure 9.** Comparison of secondary metabolite gene clusters using *Bacillus velezensis* Htq6 as the reference strain.

### 3.5. Data Mining of Transcriptome Profiles

Transcriptome analysis was conducted to investigate the difference in gene expression profiles between the experimental and control groups. As a result, 1416 genes were identified to be differentially expressed, consisting of 744 up-regulated and 672 down-regulated genes in the treated *Botrytis cinerea* relative to the wild-type strain (Figure 10A,B and Table S4). The Circos plot in Figure 11A displays the conspicuous differences in gene expression of *Botrytis cinerea* after fermentation broth treatment. Subsequently, gene ontology (GO) and KEGG pathway analyses were performed to determine the function and regulatory role of the DEGs in *Botrytis cinerea* (Figure 11B,C and Tables S5 and S6). The analyses revealed that DEGs in *Botrytis cinerea* were significantly related to carbon source utilization, energy production, cell wall, plasma membrane synthesis, and antioxidant

function. To verify these findings, qPCR was performed on six randomly chosen genes from the RNA-Seq results. The expression pattern of each gene, as quantified by qRT-PCR, was found to be in agreement with the RNA-seq data (Figure 12). Therefore, the qRT-PCR results further validated the accuracy of the RNA-Seq data.

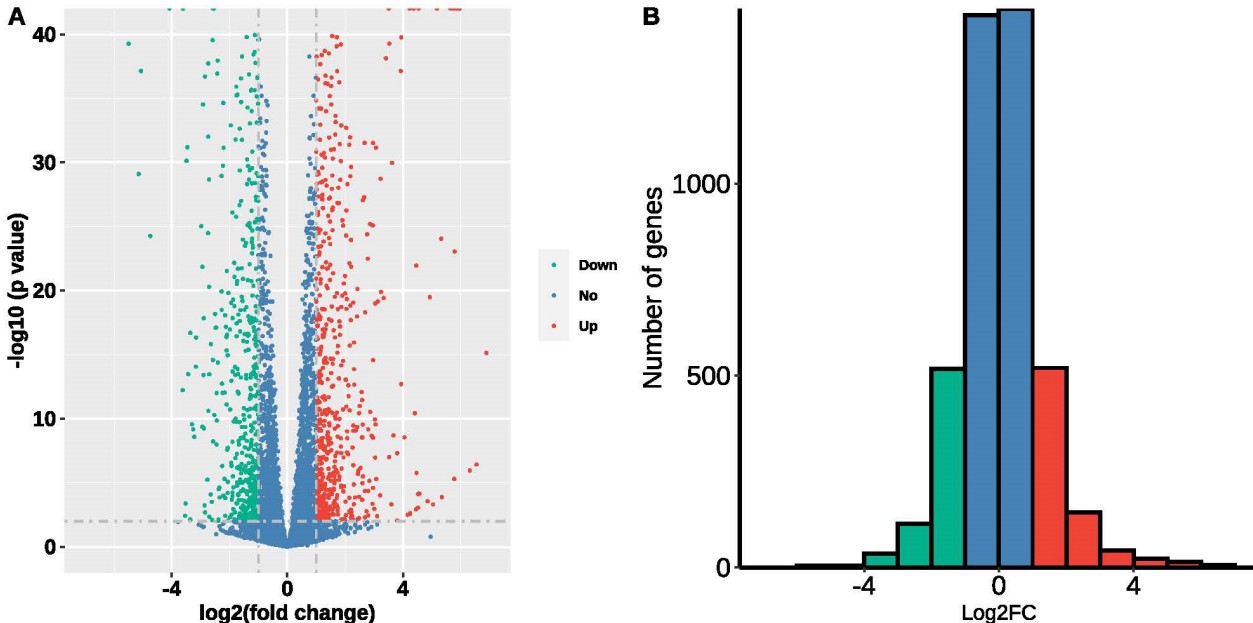

**Figure 10.** Volcano plot of gene expression patterns of *Botrytis cinerea* after fermentation broth treatment. (**A**) In the volcano plot, the *y*-axis corresponds to the mean log10 expression value (*p*-value), and the *x*-axis displays the log2 fold change value. (**B**) Histogram of the distribution of significantly different genes. The *y*-axis corresponds to the number of genes, and the *x*-axis displays the log2 fold change value. The red dots represent the significantly differentially expressed transcripts ($p < 0.05$, false discovery rate (FDR) q < 0.05) which were upregulated in *Botrytis cinerea* compared to wild type; the gray dots represent the transcripts whose expression levels did not reach statistical significance ($p > 0.05$, FDR q > 0.05) between the high and low groups.

**A**

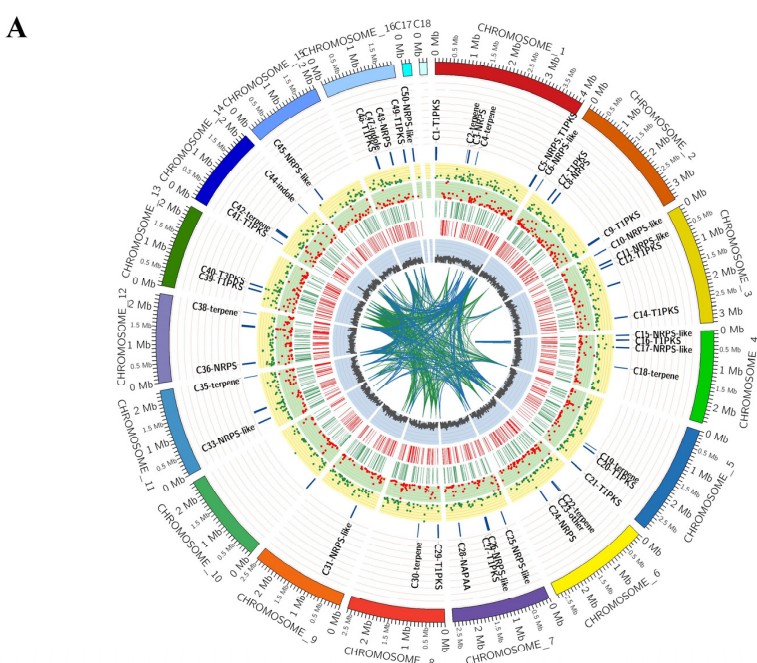

**Figure 11.** *Cont.*

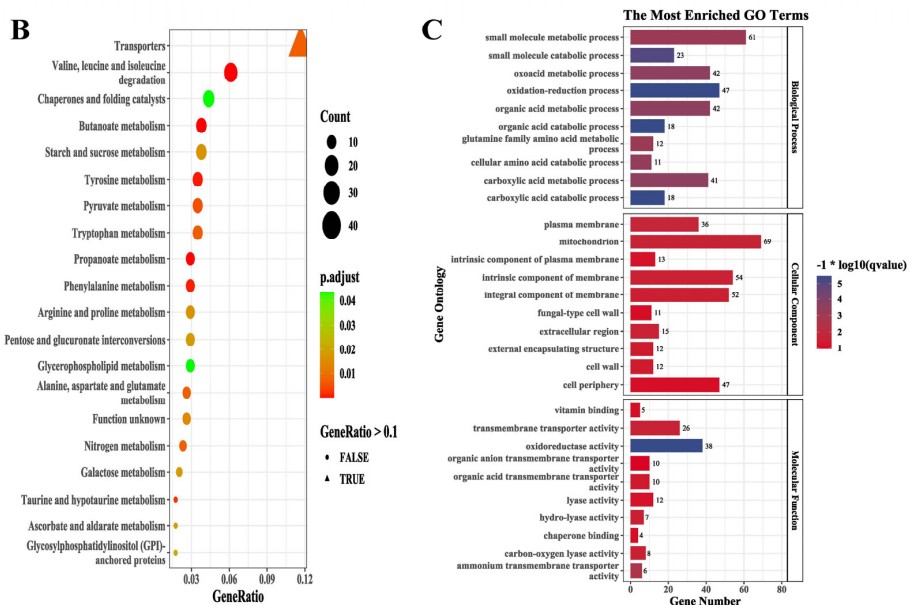

**Figure 11.** Transcriptome analysis of gene expression profiles of *Botrytis cinerea* after fermentation broth treatment. (**A**) Circos plot displaying the differences in gene expression of *Botrytis cinerea* after fermentation broth treatment. Each circle from the periphery to the core represents the following: chromosomal location; secondary metabolite gene clusters; differentially expressed genes (DEGs), with up-regulation in red, down-regulation in green; GC content; and gene duplications are shown in the center. (**B**) KEGG pathway enrichment analysis of the differentially expressed genes (DEGs) in *Botrytis cinerea* after fermentation broth treatment. The count represents the numbers of differentially expressed genes annotated in each pathway term. The Q-value is the adjusted *p*-value. Rich Factor represents the ratio of numbers of differentially expressed genes annotated in the pathway term to the numbers of all genes annotated in the same pathway. Top 20 enriched pathway terms are displayed in the figure. (**C**) Gene Ontology (GO) enrichment analysis of the differentially expressed genes (DEGs) in *Botrytis cinerea* after fermentation broth treatment. The results are summarized in three main GO categories (cellular component, molecular function, and biological process). The *x*-axis indicates the number of DEGs in a category. The *y*-axis indicates the GO term.

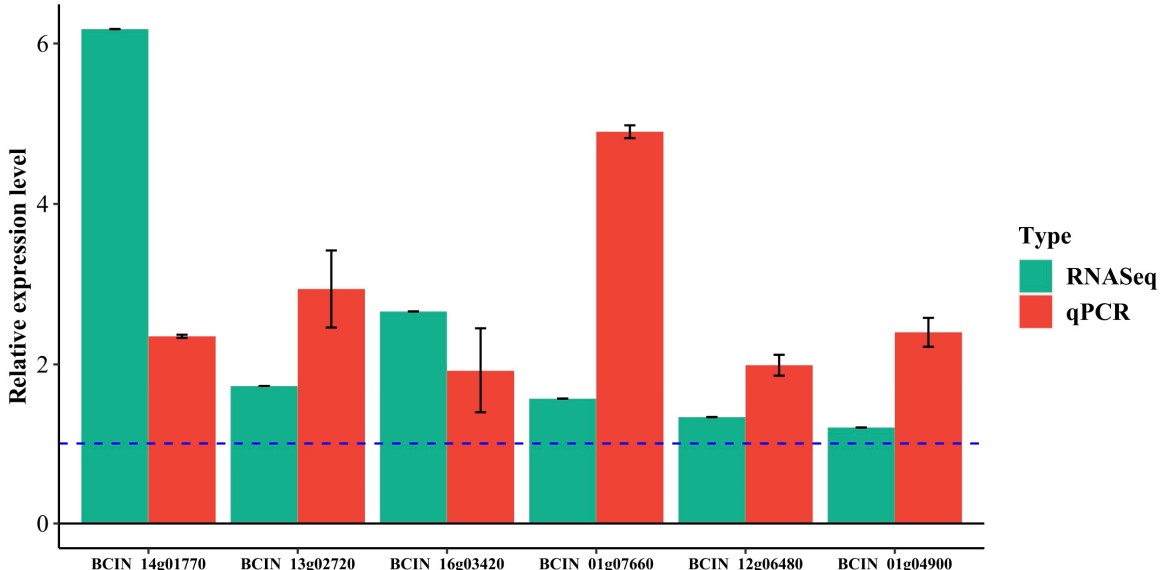

**Figure 12.** The validation of gene expression in the RNA-seq data using qRT-PCR in *Botrytis cinerea*. The blue dotted line in the figure represents Log2FC = 1.

*3.6. Effects on Genes Associated with the Carbon Source Utilization and Energy Production of Botrytis cinerea*

Studies have revealed that Htq6 has a marked inhibitory effect on the colony growth of *Botrytis cinerea*, which is likely due to its influence on the utilization of carbon sources and energy production during the growth process of *Botrytis cinerea*. KEGG pathway analysis results further highlighted that numerous differentially expressed genes were concentrated in pathways associated with carbon source metabolism, such as starch and sucrose metabolism (ko00500), galactose metabolism (ko00052), fatty acid metabolism (ko01212), and fructose and mannose metabolism (ko00051). In addition, three pathways linked to energy synthesis were also observed to be enriched with differentially expressed genes, including TCA cycle (ko00020), pentose phosphate pathway (ko00030), and glycolysis/gluconeogenesis (ko00010), which are critical for cellular ATP synthesis.

*3.7. Effects on Genes Related to Cell Wall and Plasma Membrane Synthesis of Botrytis cinerea*

The lipopeptides produced by *Bacillus* act on the fungal cell wall, causing the synthesis of the cell wall to be blocked, weakening the ability of the cell wall to act as a skeleton to maintain the cell shape and resulting in an increase in the internal osmotic pressure which leads to the cell expanding and deforming. *Bacillus amyloliquefaciens* WH1 can produce a new surfactin that can inhibit glucose polymerase and affect the synthesis of the fungal cell wall [44]. An analysis of the genome of *Botrytis cinerea* on the CAZyme online website identified 622 genes related to carbohydrate metabolism and 140 of these were found to be differentially expressed, with 72 being up-regulated and 68 down-regulated. Most of the down-regulated genes are enzymes involved in the degradation of the cell wall, such as cellulase-like cellobiose dehydrogenase, β-glucosidase, endoglucanase, and others. The up-regulated expression genes, on the other hand, involve various glycosyltransferases including α-1,3-galactosyltransferase (BCIN_10g00770), α-1,6-mannosyltransferase (BCIN_03g06440), and oxyloglucosyltransferase (BCIN_02g07230) which catalytically activate various sugar molecules, which can then be linked to different receptor molecules.

According to gene cluster analysis of secondary metabolites, Htq6 can produce surfactin and fengycin lipopeptides, which can act on the plasma membrane bilayer of pathogens and form ion channels, thereby disrupting the membrane structure, affecting the growth of pathogens and even leading to death. Moreover, fengycin has a significant influence on the stability and integrity of the plasma membrane, resulting in the outflow of intracellular substances. Of the components that make up the fungal cell membrane, phospholipids, proteins, and sugars play a vital role in maintaining a fluid membrane. Among the differentially expressed genes in the KEGG pathway, eight genes related to glycerophospholipids were found to be down-regulated. Additionally, ABC transporters, a type of transmembrane protein, are responsible for active transport of various molecules across a membrane and are key factors in drug resistance. In this case, BCIN_01g05890, BCIN_01g07660, BCIN_10g02140, and BCIN_13g02720, all ABC transporter genes, were discovered to be up-regulated, indicating that they may play a pivotal role in the membrane transport of bacteriostatic substances during fermentation.

*3.8. Effects on Antioxidant-Related Genes of Botrytis cinerea*

The treatment of *Botrytis cinerea* with *Bacillus* fermentation broth can result in an increased accumulation of intracellular reactive oxygen species (ROS). This ROS accumulation has a detrimental effect on mycelia growth, in addition to disrupting normal physiological metabolism and growth rate. From the transcriptome data, antioxidation-related differential genes were identified consisting of 4 SOD superoxide dismutase genes, 14 peroxidase genes, and 8 catalase genes. Notably, the peroxidase genes BCIN_02g06340 was up-regulated and BCIN_05g00590 was down-regulated. Additionally, catalase genes BCIN_05g04580 and BCIN_06g01180 were up-regulated, while BCIN_05g00730 and BCIN_06g04520 were down-regulated. Furthermore, through comparative analysis, 30 GST-related genes were identified in the genome of *Botrytis cinerea*; in particular, BCIN_01g04900, BCIN_12g05690,

and BCIN_14g03160 were found to be significantly up-regulated after being induced by fermentation broth. The transcriptional alteration of a large number of antioxidation-related genes indicates that the redox homeostasis in *Botrytis cinerea* cells has been disturbed.

## 4. Discussion

In the present study, we sequenced the whole genome of *Bacillus velezensis* Htq6 and compared with the genomes of *Bacillus amyloliquefaciens*, *Bacillus siamensis*, and *Bacillus velezensis*. The complete genome of the strain Htq6 was assembled, consisting of 3,888,123 base pairs with a GC content of 46.54%. Gene prediction revealed that there were 3826 protein-coding genes, which were classified into 22 functional categories of Clusters of Orthologous Groups (COGs). Among them, the largest category was general function prediction only (671 genes), followed by genes related to amino acid transport and metabolism (528 genes). In addition, a comprehensive analysis of the Htq6 strain's genome identified 146 genes related to carbohydrate active enzymes (CAZymes). CAZymes are very important proteins found in plant pathogenic fungi and bacteria, as they are essential for their growth and development [45]. These enzymes secrete substances which act on their corresponding substrates on the fungal cell wall, limiting the expansion and reproduction of pathogens by dissolving the bacteria [46]. The CAZymes can synthesize a variety of enzymes, such as chitinase, glucanase, cellulase, xylanase, pectinase, protease, etc. Not only do these enzymes help to control the growth and reproduction of bacteria that are antagonistic toward pathogens, but they are also essential for the competition between organisms in terms of nutrients and ecological sites [47].

The taxonomic identification and phylogenetic relationship of bacteria have always been the focus of research in the field of bacteriology. Initially, bacteria were classified and identified through basic morphological observation combined with some physiological and biochemical experiments or 16S rRNA sequences; however, this method is not always sufficient to differentiate between species with extremely high similarity and even DNA–DNA hybridization (DDH) could produce erroneous results [48]. Strain Htq6 is an endophytic bacterium from walnut, which was initially identified as *Bacillus amyloliquefaciens*. Taking into account the multiple evolutions of the classification status of *Bacillus amyloliquefaciens* and *Bacillus velezensis*, we sequenced the genome of Htq6, and obtained the complete genome sequences of 332 strains of *Bacillus amyloliquefaciens*, *Bacillus siamensis*, and *Bacillus velezensis* published in NCBI database. The phylogenetic analysis was conducted with *Bacillus subtilis* 168 as the peripheral strain. It was evident that 16S rRNA sequences are not enough to distinguish the 322 strains of *Bacillus*. Comparison of the complete 16S rRNA sequence revealed that a large number of strains, including *Bacillus velezensis* KCTC13012 and *Bacillus siamensis* KCTC 13613, had more than 98.5% similarity with *Bacillus amyloliquefaciens* DSM7, which is much higher than the recommended threshold of >97% or >98% for species delineation [35]. As such, 16S rRNA was not sufficient to differentiate between the three *bacillus* species. Additionally, through phylogenetic topology analysis of genomic ANI values and protein sequence similarity, the 322 strains could be grouped into three distinct clades. In 2011, Borriss proposed distinguishing FZB42 from strain groups closely related to *Bacillus amyloliquefaciens* DSM7; in order to corroborate this, a genome-wide comparative analysis and DNA–DNA hybridization were conducted. *Bacillus velezensis* FZB42 and *Bacillus amyloliquefaciens* DSM7 have been used as *Bacillus amyloliquefaciens* subsp. plantarum subsp. nov. and *Bacillus amyloliquefaciens* subsp. nov., respectively, which is a synonym of *Bacillus velezensis* sp. Nov [49]. Therefore, 58 of the 327 strains of *Bacillus amyloliquefaciens* and the Htq6 strain in this paper are incorrectly classified in terms of taxonomic nomenclature and should be reclassified as *Bacillus velezensis* (Supplementary Table S2).

By comparing and analyzing the gene clusters of antibacterial secondary metabolites of the three *Bacillus* species, they can be divided into 35 categories. Among them, cluster 11 (transAT PKS-like gene cluster) is unique to *Bacillus velezensis* and can produce difficidin, which has potent bactericidal effects and can inhibit protein synthesis [50]. Additionally, cluster 6 (transAT PKS gene cluster) is another antibacterial gene cluster shared by all *Bacil-*

*lus velezensis* strains, and is used to produce macrolactin. To date, at least 17 macrolactin species have been described and macrolactin A has been reported to inhibit Gram-positive bacteria [51]. *Bacillus velezensis* FZB42 has also been reported to produce four macrolactin substances: macrolactin A, 7-O-malonyl and 7-O-succinyl macrolactin A, and macrolactin D [52]. *Bacillus subtilis* has also been reported to produce macrolactin N, which has inhibitory effects on Escherichia coli and Staphylococcus aureus. This lactone significantly inhibits the methylase of Staphylococcus aureus peptide, and may be a new member of peptide deformylase inhibitors used as antibacterial drugs [53].

Cluster 2 (lanthipeptide) is a gene cluster of antibacterial products unique to most *Bacillus amyloliquefaciens*, which produces lanthipeptide, a class of cyclic peptide compounds containing thioether bonds, which has strong antibacterial activity against Gram-positive bacteria. By binding to the cell wall precursor lipid II, cell wall synthesis is hindered and cell death occurs [54,55]. In addition, it is also widely used as a food additive in various countries. *Bacillus amyloliquefaciens* and *Bacillus velezensis* both have their own special antibacterial active substances, which can inhibit a variety of pathogenic bacteria, which demonstrates the great potential of the two *Bacillus* in the field of biological control.

*Bacillus velezensis* Htq6 significantly inhibited the vegetative growth of *Botrytis cinerea*, and the same effect was observed with the addition of Htq6 fermentation broth in PDA medium. We hypothesized that the Htq6 could inhibit the growth of *Botrytis cinerea*, which was mainly achieved by the specific components in the fermentation broth. In order to explore the specific mechanism, transcriptome sequencing was performed. The results showed that 1416 genes of *Botrytis cinerea* were differentially expressed after being treated with Htq6 fermentation solution. After annotation, the differentially expressed genes were mainly enriched in amino acid and protein metabolism, oxidative stress, energy production, carbohydrate metabolism, and other pathways.

The genes related to energy production of *Botrytis cinerea* were seriously affected. It was found that many genes in glycolysis/gluconeogenesis pathway (ko00010), tricarboxylic acid cycle pathway (ko00020), pentose phosphate pathway (ko00030), starch and sucrose metabolism pathway (ko00500), galactose metabolism pathway (ko00052), and fructose and mannose-metabolism pathway (ko00051), which have important influences on the maintenance of energy production of organisms, were significantly changed after treated with Htq6 fermentation broth. It is known that the metabolism of starch, sucrose, galactose, fructose, and other sugars can provide substrates for the production of ATP [56]. When the energy synthesis process is blocked, the growth, reproduction, development, and other physiological functions of the organism will be severely impaired, and can even lead to cell death. In saccharomyces cerevisiae, the deletion of key genes in the glycolysis (or tricarboxylic acid cycle or pentose phosphate pathway) pathway leads to severe inhibition of vegetative growth [57]. The SNF1 protein kinase gene is crucial for the utilization of the carbon source of the pathogen, and the deletion of this gene leads to a significant decrease in the growth rate of the pathogen on nutrient-deficient medium [58,59].

The Htq6 fermentation broth treatment interfered with genes related to cell wall synthesis of *Botrytis cinerea.* All kinds of glycosyltransferases showed a marked increase, contributing to the utilization of diverse sugar sources. For example, β-1, 3-glucanase was found to be beneficial during mycelium growth, aiding in mycelium cell wall morphogenesis and conformation change, as well as in the mycelium autolysis process [60]. Furthermore, the up-regulation of genes involved in the production of cellulose, hemicellulose, and pectin was observed in *Botrytis cinerea*, which strengthens the cell wall against external damage. Thus, it is likely that *Botrytis cinerea* can reduce the damage caused by external substances by enhancing the synthesis of relevant cell wall genes. Cell walls are important for maintaining the morphology of fungal cells and protecting them from mechanical damage or osmotic stress. Studies have shown that *Bacillus* bacteriostatic substances can cause the expansion and deformation of fungal cells. Novel lipopeptide antifungal drugs can also target fungal cell walls and interfere with cell wall synthesis, thus producing a series of effects on fungal cells [61].

The genes related to cell-wall-degrading enzymes of *Botrytis cinerea* were also significantly affected. Through CAZymes gene annotations, it was found that cell-wall-degrading enzyme genes, such as fiber disaccharide dehydrogenase, β-glucosidase, endoglucanase, copper dependent lytic polysaccharide monooxygenase (LPMO), and chitinase, were down-regulated in *Botrytis cinerea*, indicating that cell-wall-degrading enzymes that act on host plants were inhibited. Fungal cell-wall-degrading enzymes play an important role in the process of pathogen infection of host plants. They can be used to not only obtain nutrients, but also to degrade host plant tissues and break through host plant cell wall defenses. Breaking through the host epidermis and degrading the host cell wall components are important for the colonization and pathogenesis of many plant pathogenic fungi. The virulence of *Alternaria alternata* in infecting Noni was related to the production of cell-wall-degrading enzymes [62].

Maintaining the stability of cellular redox homeostasis is essential for maintaining normal cellular biological functions such as proper folding of proteins and repair of DNA damage. The expression of catalase and peroxidase genes of *Botrytis cinerea* were changed. The transcriptional alteration of a large number of antioxidation-related genes implies that redox homeostasis in *Botrytis cinerea* cells has been disrupted. Meyer et al. found that the expression of glutathione transferase was significantly up-regulated in Aspergillus Niger under the stress of caspofrngin and fenpropimorph to protect the cells from damage by reactive oxygen species [63]. Among the 30 glutathione S-transferase genes, three were significantly up-regulated, while no genes were down-regulated, which may be a response of *Botrytis cinerea* to reactive oxygen species-induced injury.

**5. Conclusions**

In summary, by using genome sequencing and comparative genetics on the biocontrol bacteria Htq6, a complete genome map was obtained, and nine kinds of secondary metabolites were predicted using conserved proteins. ANI and conserved genes (gyrA or gyrB) were proposed as the basis for the classification of *Bacillus amyloliquefaciens*, *Bacillus velezensis*, and *Bacillus siamensis*. In addition, 60 strains of *Bacillus amyloliquefaciens* were found to still be misnamed and should be corrected to *Bacillus velezensis*. The sequence similarity of the secondary metabolite gene clusters of three *Bacillus* species was compared and analyzed, and the types of gene clusters were re-divided. The unique antibacterial substances of different *Bacillus* species were also elucidated. Transcriptome analysis showed that the fermentation broth of biocontrol bacteria Htq6 had significant effects on carbon source utilization and energy production, cell wall synthesis and degradation, and cell membrane and antioxidant-related genes in *Botrytis cinerea*.

**Supplementary Materials:** The following supporting information can be downloaded at: https://www.mdpi.com/article/10.3390/agronomy13061553/s1, Table S1: Oligonucleotide primers used in this study; Table S2: Genomic information of *Bacillus amyloliquefaciens* leading to reclassification as *Bacillus velezensis*; Table S3: Protein sequence of *Bacillus velezensis* Htq6; Table S4: Protein annotation of *Botrytis cinerea*; Table S5: KEGG enrichment analysis of the DEGs in the transcriptome of *Botrytis cinerea* treated with *Bacillus velezensis* Htq6; Table S6: GO analysis of the DEGs in the transcriptome of *Botrytis cinerea* treated with *Bacillus velezensis* Htq6.

**Author Contributions:** L.L.: Investigation, Methodology, Data curation, Writing—original draft. R.W.: Methodology, Writing—review and editing. X.L.: Writing—review and editing. Y.G.: Conceptualization, Methodology, Software, Data curation, Supervision, Visualization, Writing—review and editing. C.J.: Writing—review and editing. M.W.: Conceptualization, Methodology, Supervision, Visualization, Writing—review and editing, Project administration, Funding acquisition. All authors have read and agreed to the published version of the manuscript.

**Funding:** This work was supported by the Leading Industrial & Engineering Research Project of Shanxi Agricultural University (CYYL23-25), the earmarked fund for Modern Agro-industry Technology Research System (2023CYJSTX08-17), and Basic Research Program Project of Shanxi Province (20210302123404).

**Data Availability Statement:** All the genomic data have been deposited at NCBI under the BioProject accession no. PRJNA614595. The chromosomal sequence and gene annotation of *Bacillus velezensis* Htq6 can be found at GenBank with accession numbers for CP050462.

**Acknowledgments:** We are very grateful to Min Xu, Xuehan Wang, Xiaohong Lu, and Haijie Ma for their helpful scientific discussion. The transcriptome sequencing was supported by Annoroad Company (Beijing, China).

**Conflicts of Interest:** The authors declare no potential conflict of interests.

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
