# Peer review of "Characterization of a Bacillus velezensis with Antibacterial Activity and Its Inhibitory Effect on Gray Mold Germ"

_agronomy, doi:10.3390/agronomy13061553_

Round 1
Reviewer 1 Report
This is an interesting manuscript about Characterization of a Bacillus velezensis with antibacterial activity and its inhibitory effect on gray mold germ
The present work was organized logically.
However, I have some points that need to be addressed as follows.
1. In introduction you did not write anything about Botrytis why?
2. In material the authors wrote the collection of the materials what you mean by materials?
3. I see authors only isolated one isolates of Bacillus why?
4. You keep the isolate in Glycerol (what the percentage of Gelycerol?)
5. What about Botrytis pathogen?
6. If found useful, cite these recent references of Bacillus
Manzar, N.; Kashyap, A.S.; Goutam, R.S.; Rajawat, M.V.S.; Sharma, P.K.; Sharma, S.K.; Singh, H.V. Trichoderma: Advent of Versatile Biocontrol Agent, Its Secrets and Insights into Mechanism of Biocontrol Potential. Sustainability 2022, 14, 12786. https://doi.org/10.3390/su141912786
Kashyap, A.S; Manzar, N.; Nebapure, S.M.; Rajawat, M.V.S.; Deo, M.M.; Singh, J.P.; Kesharwani, A.K.; Singh, R.P.; Dubey, S.C.; Singh, D. Unraveling Microbial Volatile Elicitors Using a Transparent Methodology for Induction of Systemic Resistance and Regulation of Antioxidant Genes at Expression Levels in Chili against Bacterial Wilt Disease. Antioxidants 2022, 11,404. https://doi.org/10.3390/antiox11020404
Junior O.J.C.,Youssef K., Koyama R., Ahmed S., Dominguez A.R.,Mühlbeier D.T., Roberto S.R. 2019. Control of Gray Mold on Clamshell-Packaged ‘Benitaka’ Table Grapes Using Sulphur Dioxide Pads and Perforated Liners. Pathogens, 8, 271.https://doi.org/10.3390/pathogens8040271
Teixeira GM, Mosela M, Nicoletto MLA, Ribeiro RA, Hungria M, Youssef K, Higashi AY, Mian S, Ferreira AS, Gonçalves LSA, Pereira UP and de Oliveira AG (2021) Genomic Insights Into the Antifungal Activity and Plant Growth-Promoting Ability in Bacillus velezensis CMRP 4490. Frontiers in Microbiology, 11:618415. https://doi.org/10.3389/fmicb.2020.618415
Muhammad Imran, Abo-Elyousr KAM., Magdi Mousa, Maged M Saad 2022. A study on the synergetic effect of Bacillus amyloliquefaciens and dipotassium phosphate on Alternaria solani causing early blight disease of tomato. European Journal of Plant Pathology 162(1), pp. 63–77 https://doi.org/10.1007/s10658-021-02384-8
Moderate editing of English language
Author Response
1 In introduction you did not write anything about Botrytis why?
Response: We are grateful for the suggestion. It is necessary to include Botrytis in the introduction, as our research has applied Botrytis as an important material, and we have supplemented the relevant writing on Botrytis in the introduction.
2 In material the authors wrote the collection of the materials what you mean by materials?
Response: We apologize for the language issues in the original manuscript. For this part, we would like to introduce the source of bacterial strains and related culture conditions, so the title has been modified to "Bacterial Strains, Growth Conditions, and DNA Extraction".
3 I see authors only isolated one isolates of Bacillus why?
Response: Thanks for the reviewer's questions and concerns. We have isolated various Bacillus strain, and after conducting antibacterial tests on various pathogenic fungi, we found that Bacillus Htq6 had the best antibacterial effect and the widest antibacterial spectrum. Furthermore, basic research has shown that Bacillus Htq6 can colonize tomato plants well, and biological control agents have been preliminarily developed. Therefore, we chose the representative strain Htq6 as the tested strain and conducted in-depth discussions.
4 You keep the isolate in Glycerol (what the percentage of Glycerol?)
Response: We have indicated the glycerol usage ratio, "with glycerol-water mixtures (30 vol % glycerol)".
- What about Botrytis pathogen?
Response: We keep the Botrytis strain in glycerol-water mixtures (30 vol % glycerol).
6 If found useful, cite these recent references of Bacillus
Response: Thanks for the references provided by the reviewers. We have read these newly published papers carefully and cited them as references.
[2] MANZAR N K, A.S.; GOUTAM, R.S.; RAJAWAT, M.V.S.; SHARMA, P.K.; SHARMA, S.K.; SINGH, H.V. Trichoderma: Advent of Versatile Biocontrol Agent, Its Secrets and Insights into Mechanism of Biocontrol Potential [J]. Sustainability, 2022, 14((19)): 12786.
[12] IMRAN M, ABO-ELYOUSR K A M, MOUSA M A A, et al. A study on the synergetic effect of Bacillus amyloliquefaciens and dipotassium phosphate on Alternaria solani causing early blight disease of tomato [J]. European Journal of Plant Pathology, 2022, 162(1): 63-77.
[18] TEIXEIRA G M, MOSELA M, NICOLETTO M L A, et al. Genomic Insights Into the Antifungal Activity and Plant Growth-Promoting Ability in Bacillus velezensis CMRP 4490 [J]. Frontiers in microbiology, 2020, 11: 618415.
Reviewer 2 Report
1. The methodology should be better explained because there are procedures that are not well detailed.
2. No test showing reduction of mycelial growth of Botrytis is shown.
3. Bibliography, remove double numbering
Author Response
1. The methodology should be better explained because there are procedures that are not well detailed.
Response: Thanks for your suggestion. We apologize for not having provided sufficient detail on some of the procedures of the methodology, and we are grateful for your suggestion. In the revised version, we have elaborated on the Materials and Methods section and rearranged the content accordingly. For example, we have merged Sections 2.2 and 2.3, and changed their titles to "Genome Sequencing and Phylogenetic Analysis", while Section 2.4 on "RNA Extraction and Transcriptome Analysis" has been newly introduced.
2. No test showing reduction of mycelial growth of Botrytis is shown.
Response: Thanks for your suggestion. Bacillus amyloliquefaciens Htq6 has been demonstrated to possess prominent inhibitory activity against various pathogenic bacteria, for instance tomato gray mold, as evidenced by a publication in the journal “Fresenius Environmental Bulletin” entitled “Mutation Breeding of Bacillus amyloliquefaciens Htq6”. Therefore, in this article, we merely cited the related literature in the introduction without repeating the same result.
3. Bibliography, remove double numbering
Response: Thanks for your suggestion. According to GB/T7714-2015 format standard, we made the correct reference and removed the double numbering.
Reviewer 3 Report
Dear Author,
You have done a good job in the era where we are faced with climate change with a greater care towards our our environment and fight against food insecurity. This research work might go a long way in providing long lasting solutions. In view of it soundness for publication, I recommend you check my suggestions and comments.
A- General:
1. The results can be presented in a better manner...please try to eliminate text that are more of discussion under the result section. Report only the results. I found figures that are not referred to in the text at the end of the manuscript...is that for a particular reason?
2. The discussion section lacks enough literature backing.
3. The references might be presented in a more comprehensive manner. What criteria used for assigning DOI to some of the references?
B. Specific comments:
1. Line 3: ....plant disease and insect pest effect.
2. Lines 38-39: I recommend this sentence to suite pathogens for which biocontrol agents have been developed. The second part makes it more general in the sense that biopesticides and target a broader spectrum of pathogens.
3. Lines 41-43: rephrase this sentence, it is hard to understand. I suggest "
Although the use of agents for pest and pathogen control is a promising approach, it is still in the early stages of development and certain obstacles need to be overcome. One issue is that the agents are typically tailored to a particular species, which limits the range of organisms that can be targeted. Another challenge is the cost associated with utilizing these agents".
4. Line 56: "control" may be an appropriate word than "treatments" taking into consideration viral diseases.
5. Line 112: ...glycerol. DNA extraction can stand as a sentence on its own.
6. Line 114:...and concentration of the "genomic DNA" check the word genome in this sentence.
7. Lines 117: Extraction of the total sample RNA can be changed into (extraction of total RNA from samples)
8. Lines 117 and 123: I do have the feeling that these contents are not appropriate for the sub sections!
9. Line 175: Here, the autor failed to mention the subject to which the query sequence generated the 95% similarity.
10. Line 429: ...predicted"." and ...please delete the "."
The authors should check the length of sentences and punctuation marks.
Author Response
Comments and Suggestions for Authors
Dear Author,
You have done a good job in the era where we are faced with climate change with a greater care towards our our environment and fight against food insecurity. This research work might go a long way in providing long lasting solutions. In view of it soundness for publication, I recommend you check my suggestions and comments.
A- General:
1. The results can be presented in a better manner...please try to eliminate text that are more of discussion under the result section. Report only the results. I found figures that are not referred to in the text at the end of the manuscript...is that for a particular reason?
Response: Thanks for your suggestion, we have eliminated some text under the result section. We referred the figures in the text in the revised manuscript, thanks.
2. The discussion section lacks enough literature backing.
Response: Thanks for your suggestion, we refer to more published research articles to support our results, and fully discuss the results and supplement the citations
3. The references might be presented in a more comprehensive manner. What criteria used for assigning DOI to some of the references?
Response: Thanks for your suggestion, we apologize for the formatting error. According to GB/T7714-2015 format standard, we made the correct reference and removed the double numbering
Specific comments:
1. Line 3: ....plant disease and insect pest effect.
Response: Thanks for your suggestion, we revised this part in the revised manuscript.
2. Lines 38-39: I recommend this sentence to suite pathogens for which biocontrol agents have been developed. The second part makes it more general in the sense that biopesticides and target a broader spectrum of pathogens.
Response: We are grateful for the suggestion. We fully cite various commercial formulations developed and applied by countries such as Israel, the United States, Germany, Italy, etc. in the article, demonstrating that biocontrol is a safe, effective, and economical approach for controlling plant diseases and pests. The widespread use of these formulations attests to its efficacy and profitability.
3. Lines 41-43: rephrase this sentence, it is hard to understand. I suggest "
Although the use of agents for pest and pathogen control is a promising approach, it is still in the early stages of development and certain obstacles need to be overcome. One issue is that the agents are typically tailored to a particular species, which limits the range of organisms that can be targeted. Another challenge is the cost associated with utilizing these agents".
Response: We are sorry that we could not clearly express the meaning of the sentence, and thank you very much for your modification. we have rephrased this sentence in the revised manuscript.
4. Line 56: "control" may be an appropriate word than "treatments" taking into consideration viral diseases.
Response: We are grateful for the suggestion. As suggested by the reviewer, We revised “treatments” to “control” in the revision.
5. Line 112: ...glycerol. DNA extraction can stand as a sentence on its own.
Response: We are grateful for the suggestion. The title has been modified as " Bacterial strains, growth conditions and DNA extraction ".
6. Line 114:...and concentration of the "genomic DNA" check the word genome in this sentence.
Response: Thank you for your suggestion. We apologize for this language mistake and have taken the necessary measures to rectify it. Accordingly, we have carefully corrected this phrase throughout the manuscript in accordance with your comment, replacing the term “genome” with “genomic DNA”.
7. Lines 117: Extraction of the total sample RNA can be changed into (extraction of total RNA from samples)
Response: Thank you for your suggestion, we have made a new arrangement of the Materials and Methods.
8. Lines 117 and 123: I do have the feeling that these contents are not appropriate for the sub sections!
Response: We are grateful for the suggestion, and we agree with this opinion. We have merged 2.2 and 2.3 in the revised version, and changed the titles to “Genome sequencing and phylogenetic analysis”
9. Line 175: Here, the autor failed to mention the subject to which the query sequence generated the 95% similarity.
Response: Thanks for your careful work. We performed BLAST analysis on the 16S RNA sequencing results and the GenBank database, and the results showed that the Htq6 strains were more than 95% similar to Bacillus velezensis and Bacillus amyloliquefaciens, so we used the strains with more than 95% similarity in gyrA and gyrB gene sequences for further research. phylogenetic analysis. Through the tandem phylogenetic tree construction of gyrA and gyrB sequences, it was finally found that strain Htq6 should belong to Bacillus velezensis, which has been supplemented in Results 3.1 according to the suggestion.
10. Line 429: ...predicted"." and ...please delete the "."
Response: Thanks for your careful work. We have made the right corrections and delete the "." in the revised manuscript.